# Antimicrobial Peptides: The Production of Novel Peptide-Based Therapeutics in Plant Systems

**DOI:** 10.3390/life13091875

**Published:** 2023-09-07

**Authors:** Pragya Tiwari, Yashdeep Srivastava, Abhishek Sharma, Ramachandran Vinayagam

**Affiliations:** 1Department of Biotechnology, Yeungnam University, Gyeongsan 38541, Gyeongbuk, Republic of Korea; rambio85@gmail.com; 2RR Institute of Modern Technology, Dr. A.P.J. Abdul Kalam Technical University, Sitapur Road, Lucknow 226201, Uttar Pradesh, India; yashiids@gmail.com; 3Department of Biotechnology and Bioengineering, Institute of Advanced Research, Koba Institutional Area, Gandhinagar 392426, Gujarat, India; abhi19ind@gmail.com

**Keywords:** antibiotic arsenals, antimicrobial peptides, drug-resistant microbes, *Staphylococcus aureus*, peptide-based drugs, plant expression systems

## Abstract

The increased prevalence of antibiotic resistance is alarming and has a significant impact on the economies of emerging and underdeveloped nations. The redundancy of antibiotic discovery platforms (ADPs) and injudicious use of conventional antibiotics has severely impacted millions, across the globe. Potent antimicrobials from biological sources have been extensively explored as a ray of hope to counter the growing menace of antibiotic resistance in the population. Antimicrobial peptides (AMPs) are gaining momentum as powerful antimicrobial therapies to combat drug-resistant bacterial strains. The tremendous therapeutic potential of natural and synthesized AMPs as novel and potent antimicrobials is highlighted by their unique mode of action, as exemplified by multiple research initiatives. Recent advances and developments in antimicrobial discovery and research have increased our understanding of the structure, characteristics, and function of AMPs; nevertheless, knowledge gaps still need to be addressed before these therapeutic options can be fully exploited. This thematic article provides a comprehensive insight into the potential of AMPs as potent arsenals to counter drug-resistant pathogens, a historical overview and recent advances, and their efficient production in plants, defining novel upcoming trends in drug discovery and research. The advances in synthetic biology and plant-based expression systems for AMP production have defined new paradigms in the efficient production of potent antimicrobials in plant systems, a prospective approach to countering drug-resistant pathogens.

## 1. Overview of Global Antibiotic Resistance and Drug Discovery

The spread of antimicrobial resistance (AMR) has attained a state of ‘global emergency’ in the 21st century [1]. The statistics from the World Health Organization (WHO) have warned against the indiscriminate use of conventional antibiotics by humans in healthcare, the food sector, animal husbandry, and agriculture resulting in a high mortality of approximately 10 million deaths worldwide [2,3]. The rising frequency of antibiotic resistance and the emergence of drug-resistant diseases have a significant impact on developing countries’ economies [4,5], particularly those with limited healthcare facilities. The research initiatives have defined the ‘One Health and Global Health’ concept to address the critical themes in infectious diseases, particularly those contributing to AMR spread [6,7,8]. The interdisciplinary concept of ‘One Heath and Global Health’ provides knowledge-based information on the emergence of drug-resistant microbes and AMR evolution, the interconnected network of humans, animals, and the environment, and socio-economic factors in the spread of AMR [9,10]. Furthermore, studies have suggested that the presence of drug-resistant determinant factors in linked microbiomes alters bacterial genomes and triggers the prevalence of drug-resistant microbes [9]. An alarming rise in antibiotic resistance has been predicted by the WHO statistics suggesting 10 million mortalities every year by 2050, with most predicted causalities in the African and Asian sub-continent [11]. Besides the health of the people, the emergence of resistant microbes is worrisome in healthcare, particularly in surgical operations, transplants, and therapies for cancer treatment [12,13]. Alexander Fleming’s 1928 discovery of penicillin was a ‘phenomenal and pathbreaking discovery’ in the medical revolution, substantially expanding antibiotic arsenals for infection therapies [14]. The ‘antibiotic golden era’ was marked by streptomycin discovery and reached new paradigms with the subsequent development of sulfonamides, oxazolidinones, β-lactams antibiotics, cephalosporin, and tetracyclines, including others and the subsequently derived versions [15,16,17]. However, the efficacy and the usage of conventional antibiotics gradually declined, which is attributed to the development of drug resistance in multiple bacterial species.

The present trends in the rise of AMR can be ascribed to several causes, including indiscriminate/overuse, a lack of suitable marketing standards [18], and poor sanitation practices [19]. In the present context, methicillin-resistant *Staphylococcus aureus* (MRSA), *Klebsiella pneumoniae*, and *Mycobacterium tuberculosis* (extensively drug-resistant (XDR)) represent frequently present drug-resistant bacterial strains [20,21]. Some bacterial strains exhibited complete antibiotic resistance; for instance, colistin overuse (indiscriminate usage) to treat MDR strains (*K. pneumonia*, *A. baumannii*, and *Pseudomonas aeruginosa*) resulted in *MCR-1*-gene-mediated antibiotic resistance [22,23]. Although oxazolidinones and lipopeptides were marketed for Gram-negative microorganisms [24], staphylococcal resistance to oxazolidinones emerged subsequently [25]. The Food and Drug Administration (FDA) discontinued tigecycline and telithromycin due to their adverse effects [26,27]. *M. tuberculosis*, another bacterial strain, acquired fluoroquinolone resistance and other medicines (kanamycin, capreomycin, etc.) [28]. Daptomycin (lipopeptide) had some effectiveness in therapies but was subsequently withdrawn by the WHO [29]. The WHO produced a list of key MDR pathogens in 2017 that included third-generation cephalosporin-resistant *P. aeruginosa*, *A. baumannii*, and *Enterobacteriaceae* with an urgent necessity to tackle the emerging MDR strains [30].

The global rise in AMR further aggravated by chronic and multifactorial diseases is a threat to human and animal health and needs immediate attention. The development and characterization of novel antimicrobials from natural sources have contributed to the development of potent antimicrobials and their clinical trials in recent decades [31,32], but only a handful have shown promising outcomes. The research initiatives in this direction have explored the potential of antimicrobial peptides (AMPs) as antimicrobial agents, with key information deposited in AMP databases [33]. AMPs are emerging candidates as prospective antifungal and antibacterial therapies [34], attributed to their potential bioactivities. AMPs are a broad family of chemicals that are created as the first line of defense by multicellular organisms [35]. AMPs have been characterized as possible therapeutic alternatives to conventional therapy due to their minimal potential for resistance [36]. More than 3000 AMPs have been identified and described from multiple biological sources [37], including mammals [38], insects [39], amphibians [40], plants [41], and microorganisms [42]. In addition, AMPs are classified based on their bioactivities, namely antiparasitic, anti-tumor, antibacterial, antiviral, and anti-HIV, among others [43]. However, a detailed understanding of the AMP design and applications on the one hand and mechanisms of action on the other is crucial for creating or finding new AMPs with high biological efficacy against deadly microbial pathogens. Plants produce AMPs for a variety of purposes, including high-scale production via molecular farming and as a defense mechanism against multiple diseases (Figure 1).

The recent era has witnessed the emergence of plant systems as production platforms for AMPs as the preferred choice, attributed to synthesis in higher amounts, well-defined peptides (proper folding, glycosylation, etc.), and cost-effective peptide production, plants are favored for the manufacturing of AMPs [44]. Compared to bacterial and yeast expression systems, plant-based molecular farming is safer since there is essentially no chance of product contamination with infections that affect humans, animals, or endotoxins because plants do not have these pollutants. Substantial progress has been achieved in the production of AMPs in plant systems via synthetic biology [45]. The co-integration of synthetic biology strategies with system biology, genome engineering, and RNA interference (RNAi) for plant chassis and process optimization has yielded significant results in antimicrobial production [6,7], a plausible approach to addressing AMR. Through this thematic review, we herein present the recent trends in AMR prevalence and the prospects of AMPs as key arsenals to counter drug-resistant pathogens in a socio-economic context. In this direction, plants as production platforms and their genetic manipulation via synthetic biology comprise a sustainable approach to the production of novel and potent antimicrobials (Figure 2).

## 2. Natural Compounds and Derivatives as Potential Antimicrobial Agents

Natural compounds derived from plants, animals, and microbes are being extensively explored and studied for their antimicrobial properties [46,47,48]. Antibiotic-producing bacteria have been discovered in a variety of environmental niches, including extreme conditions (hypersaline locales) [49], plants (endophytes) [50], soil [51], marine sponges [52], and others. These antimicrobial compounds show a key potential to suppress the growth of bacteria [53] and fungi [54] and have major biotechnological/therapeutic uses. These bacteria demonstrate antimicrobial functions, and several antibiotics with high potency have been isolated from various microbial strains. Furthermore, several microbial strains were shown to suppress toxin synthesis and biofilm development [52,55]. *Lactococcus lactis* synthesizes a lantibiotic (nisin) that suppresses biofilm development in numerous species by eliminating or suppressing biofilm formation while having no negative effects on human beings. Moreira et al. demonstrated that boromycin, a macrolide from *Streptomyces antibioticus*, acts on transmembrane ion gradient and acts against non-growing and developing cells resistant to antibiotics. Plants constitute a rich reservoir of bioactive substances, and essential oils [24], triterpenoids [56], plant extracts [57], and plant proteins [58] have shown good efficacy in combating microbial infections in several animal models. Among plants, secondary metabolites with antimicrobial action are classified as phenolics, coumarins, saponins, alkaloids, terpenoids, and tannins [48,59] (Figure 3).

The enormous structural variation in chemical scaffolds of plant metabolites influences antibacterial activity, displaying multiple functions. For example, the –OH functional group found in phenolics inhibits microorganisms [59] by breaking the bacterial cell membrane and inducing cellular leakage [60], e.g., carvacrol. Eugenol is more efficient against *Listeria* and *Campylobacter jejuni* (due to double bond) than isoeugenol [61]. The antimicrobial mechanism of oxygenated terpenes comprises disruption of membranes and leakage of K^+^ ions [62]. Plant essential oils are potent antimicrobials, and limonene (1-methyl-4-(1-methyl ethenyl)-cyclohexene) is more effective than p-cymene in countering microbial infections [63]. In the case of terpenes, even minor structural changes in the location of the –OH group might drastically influence antibacterial action; for example, terpinene-4-ol at low concentrations promoted K^+^ leakage from *E. coli* cells when compared to terpineol. The capacity of oxygenated terpenes to break membranes and generate K^+^ leakage is responsible for the antibacterial effect [62]. Similarly, antibacterial activity was discovered in various plant by-products produced during food processing, in addition to other roles [64,65], due to its high phenolic content. As a result, the commercialization of by-products as antibacterials and maximal extraction would be significant in this direction.

Among the antimicrobial substances derived from animals, key examples include chitosan, lactoferrin, lactoperoxidase, lipids, and defensins [48]. Chitosan has sparked the interest of food preservationists; however, its usage is limited owing to higher and neutral pH insolubility and needs attention [66]. As acid-soluble chitosan substitutes, the chitosan derivatives have shown significant antimicrobial activity against *B. cereus*, *E. coli*, *S. aureus*, *Shigella dysenteriae*, and others [67]. Milk’s bioactive components (for example, casein) have been identified to have a variety of properties, one of which is antibacterial [68]. Coffee peels and husks [69], Bergamot peel (a by-product of essential oil) [70], pomegranate juice by-products [71], coconut husk [72], and others have antibacterial properties. Honey is a good source of bioactive substances and includes sugars, glycoproteins, etc., which have antibacterial action against MDR isolates [73]. Numerous additional bioactive compounds with antibacterial action have been identified from various natural resources, and a few notable examples are discussed (Table 1).

## 3. Antimicrobial Peptides: Novel Peptide-Based Therapeutics

AMPs are designated as a family of antibacterials that demonstrate multiple benefits including broad-spectrum anti-biofilm activity, delayed resistance development, and host immune response regulation [101]. AMPs are a class of tiny bioactive proteins that serve as a first line of defense against infections. AMPs’ mechanisms of action include immune response modulation, cell membrane disruption, and inflammation control [102] (Figure 4). AMPs are discovered in multiple niches, like soil and marine organisms, via different procedures, contributing to increased AMPs. Although 3,000 AMPs have been identified, only some have potent efficacy in clinical studies. Furthermore, the physicochemical characteristics of AMPs determine their antibacterial efficacy, absorption, and toxicity [101]. Studies have documented the safe application of synthetic AMPs in cattle, aquaculture, and as food preservatives, including the use of both natural and synthetic analogs of AMPs as antimicrobials [103,104].

### 3.1. Identification of AMPs from Nature and Properties

AMPs are biochemicals that have been conserved during evolution and are found in diverse organisms [105]. The primary characteristics of AMPs include (i) amphipathic nature (30% hydrophobic residues), (ii) positively charged molecules (+2 to +9 net positive charge) owing to positive amino acids such as lysine and arginine, and (iii) post-translational modification [106]. Furthermore, AMPs are encoded as one or more copies in biological genomes and are formed from bigger precursor molecules via post-translational modifications [106]. In the 1990s, the first AMPs were discovered in *Drosophila melanogaster* infected with bacteria or fungi [107]. A similar origin for *Drosophila*-produced human defensins has been suggested, whereas other AMPs (in *Drosophila*) exhibit homology with insect AMPs [108]. The co-evolution of AMPs in insects with other species emphasizes antimicrobial specificity, demonstrating synergistic effect and variation in specificity towards certain microbes. AMP has a broad spectrum of action and kills yeasts, bacteria, viruses, cancer cells, and fungi [109]. Plants and insects use AMPs to combat deadly microorganisms, whereas microbes use AMPs to preserve their habitats.

There are several secondary configurations for AMPs, including helices, disulfide bridges in strands, and extended/loop topologies that increase antimicrobial activity [110]. The AMPs are divided into cationic and non-cationic peptides (secretolytin) with varying structural characteristics (e.g., thanatin, penaeidins). Other characteristics include size, hydrophobicity, charge, size, self-association, etc., leading to broad-spectrum antibacterial activities [111]. Niyonsaba et al. [112] speculated that the co-evolution of AMPs may have been influenced by another characteristic of AMPs, namely the control of immune response by host cell contact. The mode of administration, target tissue, duration, dosage, and formulation, among other factors, affect how effective AMPs work against microbes. For instance, the production of AMP by the gut microbiota maintains bacterial colonization and reduces inflammation [113]. At the infection site, AMP may trigger the activation of innate immune cells and cause the induction of chemokines by a variety of cells [114]. Depending on the AMP type, it has different immunomodulatory features, including actions like those of growth factors and cytokines used to maintain immunological homeostasis [109].

The antimicrobial activity of AMPs is responsible for AMPs’ selectivity towards the target cells and is greatly influenced by their structural characteristics. Less than 100 hydrophobic and positively charged amino acid (lysine and arginine) residues make up an AMP molecule. The existence of AMPs in many biological species showed conservation over the course of evolution, and knowledge of the conserved structural features served as the foundation for creating new peptides [115]. Penaeidins, for instance, are chimeric cationic peptides that have antimicrobial action against fungi and Gram-positive bacteria. They have a chitin-binding domain (conserved), a PRP domain at the N-terminus, and a cysteine-rich domain at the C-terminus [116]. The AMP LL-37 is amphipathic and helical in shape and comprises XBBXBX patterns for heparin binding [117]. The physicochemical and structural characteristics of AMPs are another critical factor that greatly influences their toxicity toward certain cells. For instance, the AMP tachystatin was isolated from the horseshoe crab *Tachypleus tridentatus* and was antimicrobial towards fungal and bacterial strains. Tachystatin’s cytotoxic properties are thought to be caused by the presence of an amphiphilic sheet at the C-terminal end [118]. Antifungal peptides have not, however, been reported to include a conserved domain. Studies have focused on increasing AMP activity by using mutagenesis, peptide structural modification, or in silico methods. The capacity of AMPs to interact with cell membranes makes them a tumor-inducing/mitogenic agent [119] and a vector for the transport of drugs [120] and signaling molecules [121].

### 3.2. Classification and Discussion on the Structure of AMPs

In the ‘post-antibiotic era’, AMPs are emerging as potential antibacterial candidates and exhibit significant structural and functional diversity. As key components of innate immunity and the initial line of defense against microbial infections, AMPs are evolutionarily conserved in genomes. mRNA translation by ribosomes and nonribosomal peptide synthesis are the two methods that lead to the formation of AMPs in nature [122]. All living entities make and have genetically encoded AMPs from ribosomal translation, whereas non-ribosomally synthesized peptides originate from bacteria [122]. Ribosomally produced AMPs are increasingly employed as therapeutic agents for their immunological responses [35,62,109]. When AMPs are created as inactive precursors, proteases and their own expression are responsible for controlling how they are expressed [123].

According to the classification system for AMPs [124,125], most AMPs are classed as having β-sheet or α-helical structures. Despite not having a well-defined structure, the helical peptides bind with membranes to produce an amphipathic structure [126]. The two peptides from this category that have been examined as the most common are human lactoferrin and LL-97. Many AMPs are made up of β-sheet peptides, which are present in many kinds of plants, amphibians, marine invertebrates, etc. [127,128]. In the aqueous state, the β-sheet peptides have a rigid and well-organized structure and do not alter conformation as helical peptides. The antimicrobial properties of the β-sheet peptides include antibacterial, antiviral, antifungal, and anti-inflammatory effects [129]. Protegrins, defensins, and tachyplesins are some of the class’s more traditional members. The most studied β-sheet peptides are defensins, which are generated by neutrophils, epithelial cells, and macrophages as inactive precursor molecules [123,126]. These AMPs exhibit potent antibacterial action and are found in plants, invertebrates, and vertebrates [3]. Loops and extended-coil structures are included in the third class of AMPs. The extended-coil structure lacks β-sheets and -helices and is made up of amino acids such as proline, tryptophan, and arginine [35]. These AMPs work against Gram-negative bacteria by disrupting membranes and have anticancer effects as part of their broad-spectrum function [130]. Other significant examples are histatin (human saliva), tritrpticin, and indolicidin, a 13-amino acid peptide with strong antibacterial action [3,130].

### 3.3. Antimicrobial Peptides and Their Mechanism to Tackle Antibiotic Resistance

AMPs offer an advantage over conventional antibiotics since they are less prone to microbial resistance. These peptides act by disrupting membrane-bound pathogen activities and triggering the host immune system [131]. However, several issues must be resolved before AMP may be widely used as an antibiotic arsenal. Some related issues include production costs, toxicity parameters, and optimization (stability of AMPs). The development of new AMPs as therapeutic agents depends on significant research into the AMPs’ mechanism of action. Membrane-acting and non-membrane-acting AMPs are different types of AMPs. While non-membrane peptides pass across the membrane without disrupting it, membrane-acting AMPs are essentially cationic peptides that cause membrane disruption [132]. Antibacterial peptides that produce trans-membrane holes in the target membrane include LL-37 [133], defensins [134], and melittin [135]. Dermaseptin [136], buforin [137], and pleurocidin [138] are a few examples of AMPs that translocate across membranes (without producing disruption), among others. The transfer of these AMPs across cell membranes and the interruption of normal processes is their distinct mechanism of action [139]. Cell permeability and disruption are due to the cationic peptide’s interaction with the negatively charged lipopolysaccharide-containing cell membrane. Additionally, while interacting with microbial membranes, these AMPs display structural dynamics [140]. The other method of action involves blocking the creation of nucleic acids, proteins, and cell walls [141]. Additionally, AMP insertion into cell membranes is impacted by the fluidity of the membranes. The charge on the outer membrane, fluidity, etc., is essential for AMP membrane transport and influences AMP activity [142]. Additionally, certain AMPs target intracellular components by passing through the lipid bilayer and inhibit cellular processes including protein and nucleic acid production [143]. The mechanism of action of AMPs on bacterial membranes has been thoroughly studied in several models, including the carpet model [144], the toroidal pore model [145], and the barrel–stave mechanism [146].

### 3.4. AMPs as Antimicrobial Therapeutics: Clinical Validation and Trials

The AMPs highlight several benefits of their synthesis and use as antimicrobials, including high efficacy, negligible toxicity, and limited tissue accumulation. The AMPs are emerging pharmacological candidates, subject to clinical trials and validation, creating hope to expand their therapeutic applications [147]. The wider use of AMPs as therapeutics will be made possible by recent developments in peptide technology, including peptide drug conjugates, multifaceted peptides, and cell-penetrating peptides [148]. Globally, the United States produces and markets the most peptide-based medications, followed by Europe (Vicuron Pharmaceuticals and Theravance), which is principally engaged in peptide therapy research and development. Currently, over 60 peptide medications are commercially available, with additional new peptides undergoing clinical research and testing [149]. In 1999, magainin was discarded due to poor trial design, but there are still issues with peptide therapeutics that need to be resolved. One such peptide is MSI-78 which was unsuccessful in phase III clinical trials despite being effective against infections of diabetic foot ulcers. Three antimicrobial peptides (associated with indolicidin) were included in clinical trials [150]. MBI-226 was just recently launched and is currently in phase III of a clinical trial to treat catheter-related bloodstream infections [146]. By successfully reducing bacterial colonization in catheter-related infections [151] and showing antifungal activity against *Candida albicans* in guinea pigs [152], the clinical trials of MBI-226 have shown its efficiency in animal models. In a different study, Micrologix conducted phase II and III clinical trials for indolicidin-like peptides to treat acute acne and suppress MRSA [153]. AMPs and their derivatives have seen global commercial success in the treatment of infections [154]. The worldwide antimicrobial endeavors have given rise to extensive knowledge of AMPs’ effectiveness, mechanisms, safety, and other factors [155]. Additionally, greater funding for AMP research is needed to create peptide-based antibiotic drug arsenals.

## 4. Antimicrobial Peptide Production in Plants—Prospects and Advantages

Antimicrobial peptides (AMPs) comprise the first line of defense and comprise the integral component of immunity in all organisms. Recent years have witnessed increasing research on AMPs, which is attributed to their low toxicity to mammalian cells, unique action, and broad-spectrum functions [156]. The plant-based expression, also designated as molecular farming, offers a cost-effective approach to large-scale production of these therapeutic agents. Although the production of AMPs in plants is considered a prospective and emerging field of research in the present time, crucial challenges in terms of stability, function, and yield of the product account for some major concerns [157]. While plant-based expression systems are prospective as AMP production systems, an extended time is required for the optimization/generation of stable expression systems.

### 4.1. Plant Systems and Production of AMPs

Plants or animals are shielded from pathogenic attack by AMPs, serving as innate defense mechanisms [158]. There are currently more than 1700 known natural AMPs. Numerous derivatives and analogs have been created using computational systems or synthetically manufactured systems using natural AMPs as models. Because of advancements in biotechnology, it is now possible to produce proteins, peptides, and medicinal substances in large quantities using plants as bioreactors [156]. Plants produce AMPs for a variety of purposes, comprising large-scale synthesis in plants and plant protection from diseases. Due to well-organized peptides (correct folding, disulfide bond, and glycosylation) and cost-effective purification, plants are favored for the manufacturing of AMPs [43]. Additionally, compared to bacterial and yeast expression systems, plant-based molecular farming is safer since there is essentially no chance of product contamination with endotoxins or infections that affect humans or animals because plants do not have these pollutants. There are two main methods for producing AMPs in plants: steady integration of AMP genes into the plant genome, or a temporary transformation technique [159]. An illustration of molecular farming of antimicrobial peptides in plants is shown in Figure 2.

### 4.2. Plant Tissue Culture as Expression Systems

Tissue cultures, cell suspension cultures, callus cultures, and hairy root cultures are all methods used to sustain in vitro plants. Before creating a viable transgenic plant, the in vitro plant tissue culture approach is a strong tool for evaluating various expression processes [160]. This is so because plant cells or tissues may be grown in a controlled environment using regulated applications of phytohormones and culture media in in vitro cultures, which are independent of climatic variables. *Agrobacterium rhizogenes* is utilized in hairy root cultures in plant biotechnology to produce medicinal proteins and phytochemicals [161,162]. The platforms for molecular farming of antimicrobial peptides in plants are discussed. Every living plant tissue can be utilized to create a callus culture; however, young, meristematic tissues are the best sources of starting material [163]. Either direct or explant transformation of the callus and subsequent culture is used to create callus cultures that contain transgenes [164,165]. Monocotyledonous agricultural plants are typically resistant to in vitro regeneration and lack a screening platform for expression. As an alternate technique for screening stable transformants, callus cultures are used. With barley callus cultures, stable transgenics that express barley protein were created [166]. Hairy root cultures (HRCs) are the result of a plant becoming infected with the bacterium *Rhizobium rhizogenes* (initially *A. rhizogenes*). It may be generated from a variety of crops and allows the creation of extremely varied compounds [167]. HRCs have a steady biosynthetic capability and accumulate recombinant proteins in amounts comparable to those of entire plants. The stable integration of the gene of interest into the host plant genome is caused by root-inducing plasmids from *A. rhizogenes* [167]. *A. rhizogenes* was used to produce the chimeric antimicrobial peptides lactoferrampin and lactoferricin from bovine lactoferrin in tobacco hairy root cultures [44]. An appealing manufacturing strategy in terms of product consistency and the subsequent purification procedure is the expression of therapeutic proteins in in vitro plant systems under controlled circumstances [168]. ‘Next-generation human therapeutic antibodies’ are being created utilizing plant hairy root cultures [169,170,171]. Hairy root cultures of *N. benthamiana* are used to produce the tumor-targeting antibody-mAb H10 as part of an effort to discover next-generation therapeutic human antibodies [172].

### 4.3. Genetically Manipulated Plant Systems

Plant cells or tissues are used in plant molecular farming to express and process recombinant proteins or peptides with medicinal potential. This procedure is recognized as one of the most successful AMP manufacturing strategies [173]. Previously, potatoes and tobacco were commonly utilized as model plants for producing recombinant proteins, antigens, and medicines [174]. The two strategies that affect recombinant protein production in a plant are altering plant systems during synthesis (subcellular targeting, translation, and post-translational modifications) and controlling promoter-mediated transcription [175,176,177]. Constitutive production of the antimicrobial peptide cecropin P1 in transgenic potatoes resulted in plant resistance to the causal fungal pathogens of white rot and potato blight [178]. Most of the research employs constitutive expression of AMPs via ubiquitin and CaMV35S promoters [179,180,181]. Citrus Huanglongbing (HLB) and *Candidatus Liberibacter asiaticus*’ (Las) infections caused by *Xanthomonas citri* are the most damaging to citrus trees globally. Antimicrobial peptide D2A21 expression was performed to develop canker and HLB disease-resistant cultivars. Transgenic tobacco plants expressing D2A21 outperformed control plants in terms of disease resistance [182]. The LL-37 antibacterial peptide was also produced using transgenic barley. In a study, an alfalfa-plant-derived antifungal peptide was used to genetically modify potato plants [183]. Similarly, tachyplesin I, which offers defense against bacterial and fungal infections, was developed from the horseshoe crab [184]. In a different investigation, transgenic *Ornithogalum* plants expressed tachyplesin I [184]. The wild tobacco *Nicotiana* was modified for constitutive expression of the antimicrobial peptide (Mc-AMP1) from the common ice plant to examine the environmental and ecological implications of plant–microbe interactions. The outcome was that the transgenic plants displayed in-plant action against beneficial microorganisms for plants [185]. By using both direct and indirect transformation techniques, a Dermaseptin B1 recombinant antimicrobial peptide (C2-B1) was expressed in transgenic tobacco hairy root (HR) cultures. The total recombinant protein content of transgenic clones produced using direct and indirect transformation techniques and in liquid medium differs noticeably. According to the study, recombinant peptide production from the hairy root derived from the indirect transformation approach was noticeably greater [186]. *Verticillium dahlia*, an economically significant plant pathogen, is very resistant to the antimicrobial peptide BTD-S. Plants of the wild-type Columbia-0 ecotype of *Arabidopsis thaliana* expressed this peptide. In both in vivo and in vitro tests, BTD-S transgenic lines enhanced resistance toward *V. dahliae* [187]. The antimicrobial peptide PaDef (from Mexican avocado fruit) was heterologously expressed in *Pichia pastoris*, and the peptide restricted the development of *S. aureus* and *E. coli* [188] (Table 2).

### 4.4. Approaches for Transient Expression of AMPS

Due to its high expression levels, relative quickness, and high yield, transient transformation is a technology that is ideal for expressing a gene in plants and producing recombinant proteins [195]. The desired gene is expressed transiently in a plant host using a genetically designed vector, often *Agrobacterium* or a plant virus, and recombinant protein is obtained through expression within plant cells. This method allows for expression to begin within 24 h and continue for a week [196]. For temporary expression, plant viruses’ natural propensities for infection and *Agrobacterium*-mediated transformation are used. Entire plant leaves, plant suspension cells, or hairy root cultures are some of the components employed in temporary transformation [197]. Numerous peptides and proteins have been produced through temporary expression based on viral infection, including human interleukin-2, bovine lysozyme, human-galactosidase A, and bovine aprotinin, among others [198]. Using a TMV-based vector, the AMP recombinant aprotinin was created in tobacco plants. The second strategy involved employing a plant virus with high levels of expression and fast replication together with *Agrobacterium* to transfer DNA [199,200]. Heterologous proteins such as interferons, cytokines, growth hormones, etc., were produced using this method [187]. Vacuum penetration of *Agrobacterium* cultures changed tobacco plants. Even the creation of medicinal proteins or peptides for industries uses transient systems. Recombinant aprotinin was synthesized utilizing a *Nicotiana* sp. through a TMV-based vector in research. This work served as the initial illustration of the transient transformation technique for molecular farming’s generation of AMPs. The production of the avian H5N1 influenza vaccine in tobacco leaf tissues and its expression was carried out [201]. The *Agrobacterium*-mediated expression was used to synthesize the broad-spectrum AMP protegrin-1 (PG-1) in *Nicotiana tabacum*. According to Patio-Rodrguez et al. [202], this peptide was physiologically active against several bacterial and fungal species. The fruits of *Raphanus sativus* were resistant to bacterial spot disease by the induced production of SP1-1, an AMP targeted to the apoplast [203].

The use of plant cell suspension cultures requires consistent and specified conditions, which provide superior quality monitoring with a suitable medium, particularly during continuous manufacturing operations [195]. The scalability of plant cells, which is lower than that of full plants, is the main limitation of plant cell cultures. Additionally, it is not always possible to apply techniques that have been refined for plant cell cultures to actual plants [204]. Therefore, more screening and translational studies are further required. To establish a connection between plant cells and the scalability of complete plants, a plant cell pack (PCP), usually referred to as “cookies”, is created. The foundation of this platform is plant cell cultures without a liquid medium. Because of this, plant cell packs offer a viable platform for metabolic engineering, synthetic biology, and typical recombinant protein expression techniques that need high throughput screening of a variety of constructs for effective product development [205]. Human antibody chain proteins were molecularly farmed using PCP technology [206]. Rademacher developed a technique for the development and growth of a plant cell pack in 2020. A limitation to the industrial use of plant-based expression systems is the limited quantity of transiently expressed recombinant proteins generated via entire plants or plant cells. PCPs offer a quick and high-throughput screening approach with increased recombinant protein expression. In cell suspension cultures, namely *A. thaliana*, *Catharanthus roseus*, *N. tabacum*, etc., PCPs are effective. Due to fewer host proteins and secondary metabolites than in leaf-based expression systems, the approach also facilitates product purification. *Agrobacterium tumefaciens*-infused plant cell packs (PCPs), which are three-dimensional, porous aggregates of plant cells, were developed as an expression method [207]. For plant species like *Nicotiana* spp. and *D. carota*, these PCPs are more suited and effective than the transient expression approach in a liquid plant cell culture [206]. This uses both fast-growing cell suspension cultures and protein expression assays to understand vector replication efficiency detected as red and green fluorescent proteins [207]. The high-throughput screening of recombinant protein expression has been aided by the development of PCPs. The newly automated technique offers an affordable platform with more sample detection [208]. The same team of researchers has developed a technique for automatically transforming plant cell packs [208].

## 5. Computational Resources and Antimicrobial Research

The advancement of computational biology has made it easier to create vaccines and medicines with greater effectiveness. The development of antibacterials has centered on a combinational chemistry approach combined with computational modeling of the 3D structure of the antibacterial using energy minimization and other methods [209]. This approach aims to identify an effectively docked compound to a virulent gene, restricting the growth of the pathogen. The capacity to store proteomics/genomics findings and retrieve biological information has solved the issues connected with combating drug-resistant diseases. This has been made possible by the decoding of biological genomes and genome analysis using automated techniques [210]. The advancements in computational biology have made it easier to create vaccines and medicines with greater effectiveness. The antibacterial development has centered on combinational chemistry combined with computational modeling of the 3D structure of the lead compound using energy minimization and other techniques [209]. This approach aims to identify an effectively docked compound to a virulent gene, restricting the growth of the pathogen. The capacity to store proteomics/genomics findings and retrieve biological information has solved the issues connected with combating drug-resistant diseases. This has been made possible by the decoding of biological genomes and genome analysis using automated techniques [210]. Several bioinformatics databases and tools have also been created to give data on AMPs, information on microorganisms, and in silico methods to examine interactions between human pathogens.

Most of the research into the creation of antibacterial drugs has, up to this point, concentrated on the synthesis of lead candidates via inhibiting the synthesis of nucleic acids, limiting the biosynthesis of peptidoglycan, and blocking crucial metabolic pathways [210,211]. Through the identification of plasmid genes that inhibit the pathway, sequencing of genomes, and metabolic pathway reconstitution, computational biology has made a substantial contribution to drug development efforts [210]. To fully integrate bioinformatics and biochemical analysis for the generation of prospective drug candidates, key studies are still required. The advances in bioinformatics have contributed to our understanding of the existence of surface antigens and pathogenic genes. To create effective antimicrobial candidates for drug discovery and development, a deeper knowledge of microbial dynamics is required.

## 6. Bottlenecks in Research on AMPs as Antimicrobial Therapeutics

The substantial research and progress on AMPs, has necessitated the exploration of natural products to develop novel and potent candidates as antimicrobial agents. More than 3000 AMPs have been found and described to date [212], yet the bulk of them are unsuitable for treating humans (in their natural state) and many of them failed in clinical trials. AMPs, including telavancin, dalbavancin, vancomycin, daptomycin, oritavancin, and gramicidin D, have received FDA approval and are being marketed for use in medicine [213]. Many peptide-based therapies in the market function by activation or blocking of receptors, and others include pathway inhibitors and peptides that act on membranes. For usage as drug molecules, peptide-based treatments must be stable; most peptides authorized by the FDA exhibit stability in vivo [214]. Designing antimicrobials from bacteria and their biochemical aspects, combining AMPs with drugs for better function, and developing in vitro systems to study the antimicrobial effects are all necessary for creating synthetic AMPs [213]. Antimicrobial agents’ broad-spectrum action is caused structurally by the union of a clearly defined cationic region with a hydrophobic surface area, yet the rigidity of AMPs’ structure impairs their abilities to carry out the functions [128]. The main obstacles to AMP usage are its effectiveness and toxicity. Since some AMPs have in vivo side effects, they are exclusively employed in topical applications [215]. Additionally, the presence of serum and excessive salt levels might impact the antibacterial effects [144]. The vulnerability of AMPs to host proteolytic enzymes, which break down AMPs and influence their stability and pharmacokinetics, is another significant issue [216]. As a result, several AMPs are only recommended for use in tropical climates and are not recommended for oral administration. Current research has shown a more varied and complicated mechanism of AMPs, emphasizing a larger and multifaceted approach to combating microbial infections. In addition to their biological characteristics, additional elements such as in vivo stability [214], together with the resulting side effects and production costs, must be taken into consideration. To create AMPs with enhanced function, it will be crucial to have a thorough understanding of structure–function studies of AMPs and their effectiveness range.

## 7. Commercial Potential and Prospects of AMPs as Therapeutic Agents

AMPs are being developed as next-generation antibiotics to treat a variety of microbial illnesses, notably MDR strains, despite the numerous projected limitations. The tremendous therapeutic potential of AMPs, both natural and synthesized, as novel antimicrobials is highlighted by their multiple mechanisms of action. Our understanding of the structure and characteristics of AMPs has recently progressed, which is attributed to scientific advances, but there is still a long way to go before these therapeutic options can be fully utilized. Additionally, modified AMPs target or show specific antimicrobial activity with enhanced characteristics [217]. These modified AMPs include hybrid peptides, peptide mimetics, immobilized peptides, and peptide conjugates. These innovative peptides have potential uses in both industry and medicine, including as arsenals against antibiotic-resistant bacterial strains and for food preservation. AMR is on the rise all over the world, and AMPs have the potential to combat drug-resistant organisms, emphasizing the development of new antimicrobial options for viral, fungal, and bacterial illnesses. Short synthetic AMPs are frequently used to combat MDR microbial strains, and significant progress has recently been achieved in the design and synthesis of short peptides with increased effectiveness and decreased toxicity. The discovery of AMPs offers promising opportunities to replenish the depleting antibiotic pipeline and combat the global rise in AMR.

## Figures and Tables

**Figure 1 life-13-01875-f001:**
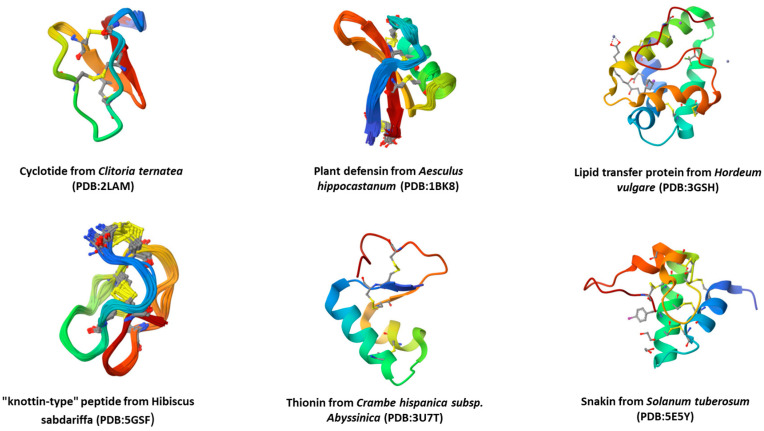
Representative examples of antimicrobial peptides produced in plant systems as effective antimicrobials (source: 3-dimensional structures downloaded from RCSB Protein Data Bank, https://www.rcsb.org/ (accessed on 20 August 2023).

**Figure 2 life-13-01875-f002:**
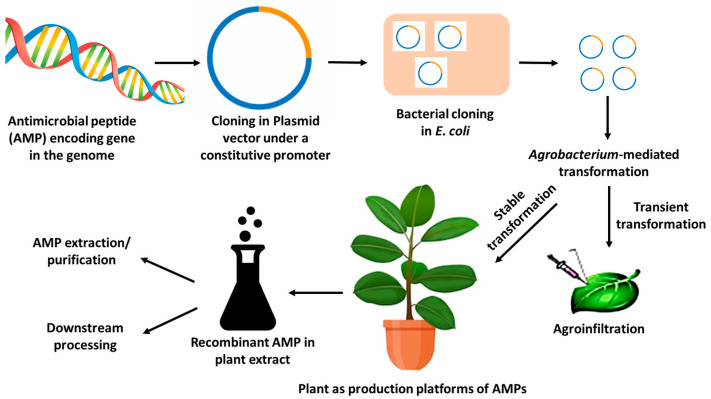
Plant systems as ‘production platforms’ for molecular farming of novel and potent antimicrobial peptides.

**Figure 3 life-13-01875-f003:**
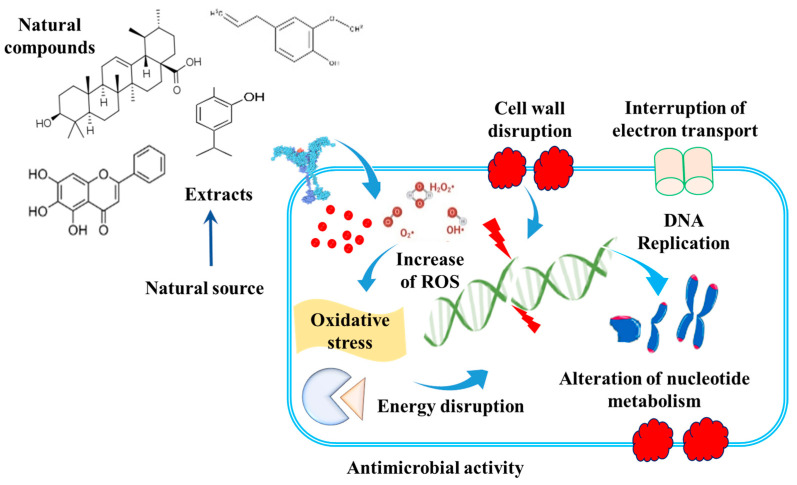
Significance of natural compounds and derivatives as potential antimicrobial agents.

**Figure 4 life-13-01875-f004:**
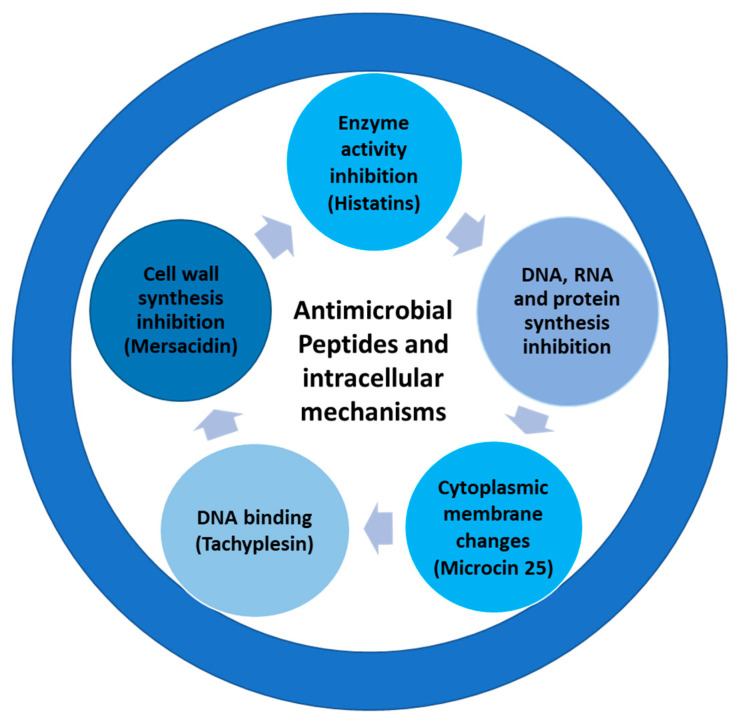
Antimicrobial peptides: novel peptide-based therapeutics and their mechanisms.

**Table 1 life-13-01875-t001:** Key examples of phytomolecules as potent antimicrobials, their bioactive constituents, and modes of action against pathogenic microbes (modified from Tiwari et al. [7]).

S. No.	Classification	Key Examples	Mechanism of Action	Reference
1.	Plant-derived compounds			
	Phenolics	Thymol, Carvacrol	The hydroxyl group increases the Membrane disruption and Leakage of cellular contents	[60]
	Flavonoid compounds	Catechins	Antibacterial against *Shigella*, *Vibrio*, and *Streptococcus mutans*	[74,75]
	Hydroxylated phenols	Catechol and pyrogallol	Antibacterial against *Corynebacterium xerosis*, *Pseudomonas putida*, and *P. pyocyanea*, and antifungal (catechol) against *Penicillium italicum* and *Fusarium oxysporum*	[76]
	Polyphenol (3,5,4′-trihydroxystilbene)	Resveratrol	Antifungal against *Plasmopara viticola*, *Sphaeropsis sapinea*, and *Pyricularia oryzae*. In *C. albicans*, resveratrol penetrates the cell membrane and causes apoptosis. Antibacterial against *M. tuberculosis*, VRE, *S. typhimurium*, and MRSA	[77,78,79,80]
	Essential oil from *Salvia fruticosa*	----	Inhibition of Efflux pump in *Staphylococcus epidermidis* (clinical isolates)	[81]
	Quinones from Juglans and Plumbago	Juglone and plumbagin	Antibacterial against *S. aureus* by increasing membrane permeability and restricting the formation of cell wall	[82]
	Essential oil from *Chenopodium ambrosioides*	----	Efflux pump Tet(K) inhibition in *S. aureus* IS-58	[83]
	Alkaloid	Capsaicin	Efflux pump NorA inhibition in *S. aureus* SA-1199B	[84]
	Anthraquinone from *Hypericum perforatum*	Hypericin	Antimicrobial activity against methicillin-resistant and methicillin-sensitive *Staphylococcus*	[85]
	Alkaloid	Catharanthine	Efflux pump inhibition in *P. aeruginosa*	[86]
	Dimeric Phenylpropanoids from *Styrax japonica*	Lignans Styraxjaponoside C	Antifungal against *C. albicans* showing membrane-active mechanisms	[87]
	Flavonoid	Baicalein	*S. aureus* SA-1199B NorA efflux pump inhibition	[88]
	Triterpenoids	Ursolic acid and derivatives	inhibition of efflux pump AcrA/B, MacB, TolC and YojI in MDR *E. coli* (KG4)	[89]
2.	Plant by-products in food processing			
	Green husks of walnuts	----	Antibacterial against *B. subtilis*, *S. aureus* and *B. cereus*	[90]
	Grape pomace	Phenolics	Growth is hampered in *S. aureus*, yeasts, and *Salmonella* sp.	[91]
	Bergamot peel, an essential oil by-product	Chlorogenic acid	Antibacterial against *B. subtilis* and food-borne *E. coli*, *S. enterica*	[70]
	Pomegranate fruit peel extracts	Phenolic constituents	Hampered growth in *S. aureus*, *Y. enterocolitica*, *L. monocytogenes*, etc.	[92]
	Pomegranate rind	Tannins	Antimicrobial against *L. monocytogenes* modify microbial cell membranes and impair cell homeostasis	[93]
	Coconut husk	Tannins and Phenolic constituents	In *L. monocytogenes*, and *V. cholera*, growth is hampered	[72]
	Olive juice powder and olive pomace	Phenolic compound (oleocanthal)	Antimicrobial against *L. monocytogenes*, *S. aureus*, and *E. coli*	[94]
3.	Animal-origin compounds			
	Chitosan	Polycationic biopolymer compound	Antibacterial towards *L. monocytogenes*, *B. cereus*, *S. aureus*, and others	[64]
	Milk-derived substances (casein and whey proteins)	----	Antibacterial/antifungal against *Helicobacter*, *Listeria*, *Salmonella*, *Staphylococcus*, *E. coli*, yeasts, and filamentous fungi	[95]
	Lysozyme	Bacteriolytic enzyme	Lysozyme hydrolyzes the β-1, 4 linkages between N-acetylmuramic acid and N-acetylglucosamine in the peptidoglycan of the microbial cell wall	[64]
4.	Antimicrobials of bacterial origin			
	Bacteriocin	Nisin	Growth is hampered in Gram-positive and spore-producing bacteria in food	[96]
	Reuterin	β-hydroxypropionaldehyde	Antimicrobial towards foodborne pathogens	[97]
5.	Antimicrobials from algae and mushrooms			
	Phlorotannins from marine brown algae	----	Antimicrobial towards *S. aureus*, *Salmonella* spp., etc.	[98]
	Grifolin, and pleuromutilin from macrofungi	----	Antimicrobial activity against *S. aureus*, *B. cereus*, *L. monocytogenes*, *E. coli*	[99]
	Fatty acids, β-carotene-linoleic acid, flavonoids from *Agaricus* spp.	----	Antimicrobial towards *Micrococcus luteus*, *B. cereus*, etc.	[100]

**Table 2 life-13-01875-t002:** Representative examples of genome editing in biological organisms for applied biotechnologies.

S. No.	Genome- Editing Tool	Biological System	Method of Genome Editing	Applications	Reference
1.	Cas9/sgRNA system	*Cucumis* *sativus*	eIF4E (eukaryotic translation initiation factor 4E) gene was targeted by Cas9/sgRNA construct	Plant resistance to viruses	[189]
2.	RNA interference (RNAi)	*Synechocystis* sp.	CRISPR-RNA (crRNA) in *Synechocystis* sp. degrades target mRNA	To address drug resistance in microbes	[190]
3.	Genetic engineering	*Arabidopsis* *thaliana*	Cht1 signal peptide (Cht1SP)- thanatin(S)-GFP construct was introduced in the plant	Antifungal and antibacterial activity of Thanatin(S), antimicrobial peptide	[191]
4.	Metabolic engineering/ overexpression	*Poplar* spp.	MsrA2 (antimicrobial peptide) overexpression in the plant	Improved pathogen resistance	[192]
5.	*De novo* designing of AMP and engineering in plant	*Nicotiana* *benthamiana*	SPI-I (AMP) was designed and introduced in plant system	Antimicrobial function against pathogens	[193]
6.	Plant transformation	*Oryza sativa*	Plant transformation with cecropin A gene	Fungal and bacterial pathogen resistance	[194]

## Data Availability

Not applicable.

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
