# Peer review of "Antimicrobial Peptides: The Production of Novel Peptide-Based Therapeutics in Plant Systems"

_life, 2023, doi:10.3390/life13091875_

Round 1

Reviewer 1 Report

Recommendation: Major revision

Comments:

This manuscript, titled "Antimicrobial Peptides: Production of Novel Peptide-Based Therapeutics in Plant Systems," provides a comprehensive overview of the current sources and classifications of antimicrobial peptides, as well as their mechanisms of action and resistance. Although authors put a lot of effort into collecting date and literature for to elucidate current progress, the manuscript needs major changes/corrections before acceptance for publication. Some comments are useful:

1. Chapter 4 lacks an introduction to the following four sections.

2. The “AMP databases” mentioned in line 91 do not indicate the source, which may be important for some readers.

3. It may be useful to show structural information, such as phytomolecules and antimicrobial peptides mentioned in the manuscript.

4. In order to enhance the legibility of the article, the depiction of the mechanism of action of antimicrobial peptides can be presented in a visual representation.

Author Response

On behalf of all the authors, I wish to submit the revised manuscript “Antimicrobial Peptides: The Production of novel peptide-based Therapeutics in plant system” for possible consideration in MDPI Life, a renowned journal of repute in the life science field.

The authors acknowledge the valuable/critical comments made by the esteemed reviewers for the improvement of the manuscript, All the comments are addressed point-by-point and details are enclosed.

Please refer to the revised manuscript for changes, In addition, Table 1 is considerably revised and Table 2 is prepared for the sake of avoiding plagiarism. Minor English language errors, punctuation and grammer is addressed.

Comments:

This manuscript, titled "Antimicrobial Peptides: Production of Novel Peptide-Based Therapeutics in Plant Systems," provides a comprehensive overview of the current sources and classifications of antimicrobial peptides, as well as their mechanisms of action and resistance. Although the authors put a lot of effort into collecting date and literature for to elucidate current progress, the manuscript needs major changes/corrections before acceptance for publication. Some comments are useful:

Response: We are grateful for the useful comments made for the improvement of the manuscript.

  1. Chapter 4 lacks an introduction to the following four sections.

Response: We thank the esteemed reviewer for the important point. In chapter 4- an introduction “Antimicrobial peptide production in Plants-Prospects and Advantages” has been added, please refer to the revised manuscript for changes.

  1. The “AMP databases” mentioned in line 91 do not indicate the source, which may be important for some readers.

Response: The source of AMP Databases is added.

  1. It may be useful to show structural information, such as phytomolecules and antimicrobial peptides mentioned in the manuscript.

Response: The authors abide by the suggestions. Figure 1. Representative examples of Antimicrobial peptides produced in plant systems as effective antimicrobials depicts the 3dimensional structures of key plant antimicrobial peptides, please refer the revised manuscript.

  1. In order to enhance the legibility of the article, the depiction of the mechanism of action of antimicrobial peptides can be presented in a visual representation.

Response: We are thankful for this comment. Figure 4. Antimicrobial peptides: novel peptide-based therapeutics and their mechanisms has been drawn to include key mechanisms of antimicrobial peptides. Please refer to the manuscript for changes.

Reviewer 2 Report

Dear Athors,

The manuscript deals with an interesting topic, offering interesting information. The review is well documented. The manuscript is well-written and I do not have any comments. I recommend to accept in present form.

Author Response

Dear Authors,

The manuscript deals with an interesting topic, offering interesting information. The review is well documented. The manuscript is well-written and I do not have any comments. I recommend to accept in present form.

Response: The authors greatly appreciate and thank the esteemed reviewer for the positive and encouraging comment on the submitted manuscript. We hope to put in our best efforts in future too, thank you

Reviewer 3 Report

The review "Antimicrobial Peptides: the production of novel peptide-based therapeutics in plant system" is well written, well structured, and comprehensively addresses issues related to the study, production, and use of antimicrobial peptides. The topic of the review itself is quite popular and relevant. I have no key remarks on the content of the presented work.

However, there are ambiguities in the text:

Line 230: The expression "positive charge (+2 to +9 net positive charge)" is not entirely clear; perhaps it is worth clarifying or adding units of measurement (and also further in the text in similar cases).

Line 491: There is a factual or grammatical error in the expression "Raphanus sativus tomato fruits". The entire sentence needs to be reformulated.

Line 569: "With substantial research and progress on AMPs, which are a natural part of the innate immune response. Exploring chemicals from natural sources has become necessary because of pipeline drying and the overproduction of antibacterial arsenals." These two sentences seem completely meaningless to me and should be rewritten or replaced.

After minor edits, the review may be published.

Author Response

The review "Antimicrobial Peptides: the production of novel peptide-based therapeutics in plant system" is well written, well structured, and comprehensively addresses issues related to the study, production, and use of antimicrobial peptides. The topic of the review itself is quite popular and relevant. I have no key remarks on the content of the presented work.

Response: The authors express their gratitude for the positive remarks of the esteemed reviewers.

However, there are ambiguities in the text:

Line 230: The expression "positive charge (+2 to +9 net positive charge)" is not entirely clear; perhaps it is worth clarifying or adding units of measurement (and also further in the text in similar cases).

Response: line 230, the sentence was rewritten and the details of positive amino acid residues, arginine and lysine were added, please refer to the changes.

Line 491: There is a factual or grammatical error in the expression "Raphanus sativus tomato fruits". The entire sentence needs to be reformulated.

Response: We are thankful for rectifying our mistake. The sentence did not made any sense and was rewritten for the sake of clarity.

Line 569: "With substantial research and progress on AMPs, which are a natural part of the innate immune response. Exploring chemicals from natural sources has become necessary because of pipeline drying and the overproduction of antibacterial arsenals." These two sentences seem completely meaningless to me and should be rewritten or replaced.

Response: The 2 sentences were corrected in the revised manuscript.

After minor edits, the review may be published.